# A coupled human-Earth model perspective on long-term trends in the global marine fishery

E.D. Galbraith[1,2,3], D.A. Carozza[3,4] & D. Bianchi[5]

The global wild marine fish harvest increased fourfold between 1950 and a peak value near the end of the 20th century, reflecting interactions between anthropogenic and ecological forces. Here, we examine these interactions in a bio-energetically constrained, spatially and temporally resolved model of global fisheries. We conduct historical hindcasts with the model, which suggest that technological progress can explain most of the 20th century increase of fish harvest. In contrast, projections extending this rate of technological progress into the future under open access suggest a long-term decrease in harvest due to over-fishing. Climate change is predicted to gradually decrease the global fish production capacity, though our model suggests that this is of secondary importance to social and economic factors. Our study represents a novel way to integrate human-ecological interactions within a single model framework for long-term simulations.

[1] Institució Catalana de Recerca i Estudis Avançats (ICREA), Pg. Lluís Companys 23, Barcelona 08010, Spain. [2] Department of Mathematics, Institut de Ciència i Tecnologia Ambientals (ICTA), Universitat Autònoma de Barcelona, Barcelona 08193, Spain. [3] Department of Earth and Planetary Sciences, McGill University, Montreal, Québec H3A 0E8, Canada. [4] Department of Mathematics, Université du Québec à Montréal, Montréal, Québec H3C 3P8, Canada. [5] Department of Atmospheric and Oceanic Sciences, University of California, Los Angeles, California 90095, USA. Correspondence and requests for materials should be addressed to E.D.G. (email: eric.galbraith@uab.cat).

The global capture of wild marine 'fish' (including edible invertebrates, as well as true fish) appears to have peaked in the 1990s, according to a recent reconstruction[1] (Fig. 1a), despite continued increases in the effort expended by the global fishery[2] (Fig. 1b). The increasingly intense fishing pressure has driven marine fish biomass to a fraction of its pristine state (Fig. 1c), a depletion that is widely believed to be limiting fishing yields in many parts of the world[2–4]. These historical trends reflect the interplay of ecosystem dynamics, the demand for fish, the cost of fishing, improvements of fishing technology and climate change[1,3,5–7], all of which will play a role in future[8–10], but whose relative roles have been difficult to formally assess.

Here we quantitatively address the multiple influences in the global fishery, and their interactions, by explicitly including human activity within an Earth System modelling framework, and using simulation protocols typically used for climate simulations. We use the BiOeconomic mArine Trophic Size-spectrum (BOATS) model, a bio-energetically constrained macroecological-life-history fish model that is coupled directly with an economic model[11,12] (Methods, see Supplementary Methods for details). The BOATS model builds on prior works that took regional[13], species-specific or unidirectional coupling approaches[14,15] by introducing a comprehensive two-way coupling of human and natural components of the system, using relatively simple but well-founded predictive principles applicable to multi-decadal timescales. Primary production by phytoplankton and seawater temperatures are used to predict the growth and reproduction of fish, by determining the energy available to the trophic web[16] and the metabolic rates of size-structured fish populations[17]. The fish harvest is determined by local-biomass density and interactive fishing effort, which evolves independently in each grid cell over time. As a first-order approximation, we assume that individual fishermen are rational and profit-seeking, as generally borne-out by observations[18], and that there are no property rights (Open Access (OA)), which is the expected outcome in unmanaged fisheries. The current model version does not simulate artisanal or recreational fisheries explicitly, but only industrial fisheries, which account for >80% of global harvest and most of the 20th century trend[1]. Although the OA dynamic is not representative of the ~40% of fisheries where management has recently become relatively effective[19], it was a reasonable approximation of global fisheries throughout most of the 20th century[20] and remains reflective of at least half of fisheries today, particularly in low-income countries[21,22]. Poorly constrained ecological model parameters are calibrated using Monte Carlo-based approximate Bayesian computation (Methods, Supplementary Methods) by comparing simulated with observed catch and biomass, aggregated at the scale of Large Marine Ecosystems (LMEs). The use of globally distributed LMEs allows the assessment of model performance across the full range of extant environmental conditions[12]. We show results for an ensemble of five different combinations of parameter values that provide realistic solutions, while representing a broad range of parameter uncertainty.

Because the model resolves the dynamics of the global fishery as a function of economic and environmental conditions, historical 'hindcast' simulations can be conducted to attribute the historical harvest trend to possible drivers on the coupled system, analogous to the approach taken with coupled ocean-atmosphere models to identify the role of greenhouse gases in historical climate[23]. Given our simple but inclusive conceptual framework, the external factors influencing the long-term development of the wild capture fishery are climate, the price of fish, the cost of fishing and the technology-dependent ability of fishermen to catch fish for a given amount of fishing effort. We

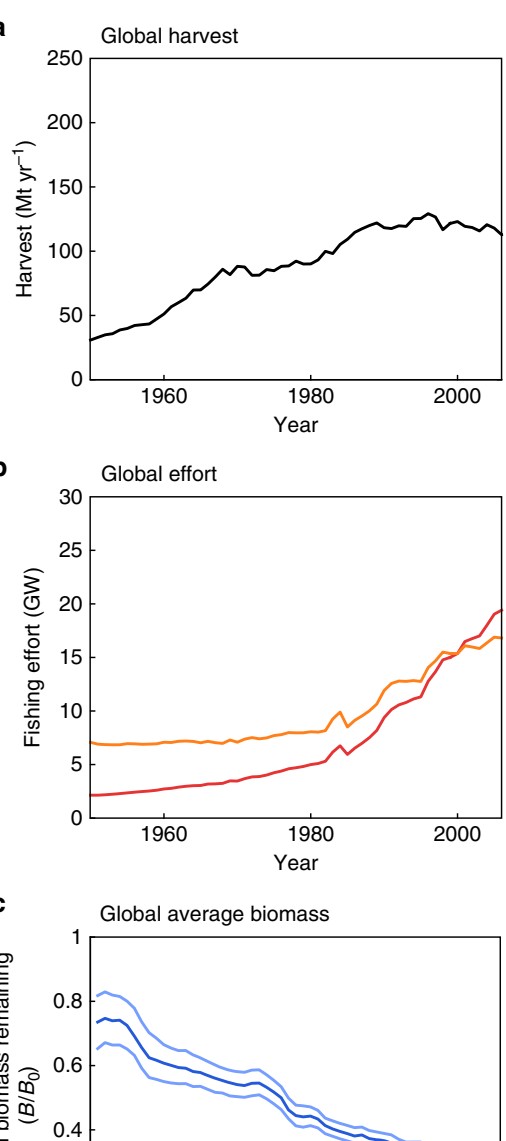

**Figure 1 | Global historical time-series characterizing the global wild capture fishery.** (**a**) Reconstructed global fish harvest[1], including illegal, unreported and underreported catches. (**b**) Estimated nominal effort (orange) and effective effort assuming, conservatively, an increase in efficiency of 2.4% per year (red) (ref. 2). (**c**) Biomass as a fraction of pristine biomass from selected stock assessment data, as estimated by ref. 4.

first consider the ability of these factors to explain broad features of the observed 20th century trend of fish harvests, and then consider their possible roles in the future of the wild capture fishery.

Our results show that the historical increase of wild fish harvest, aggregated at the global scale, can be reproduced to first order by a hindcast that includes a moderate rate of increase in the technology-dependent ability to catch fish over time, assuming open access. We also show that, were it to continue, this same rate of technological progress would dominate the

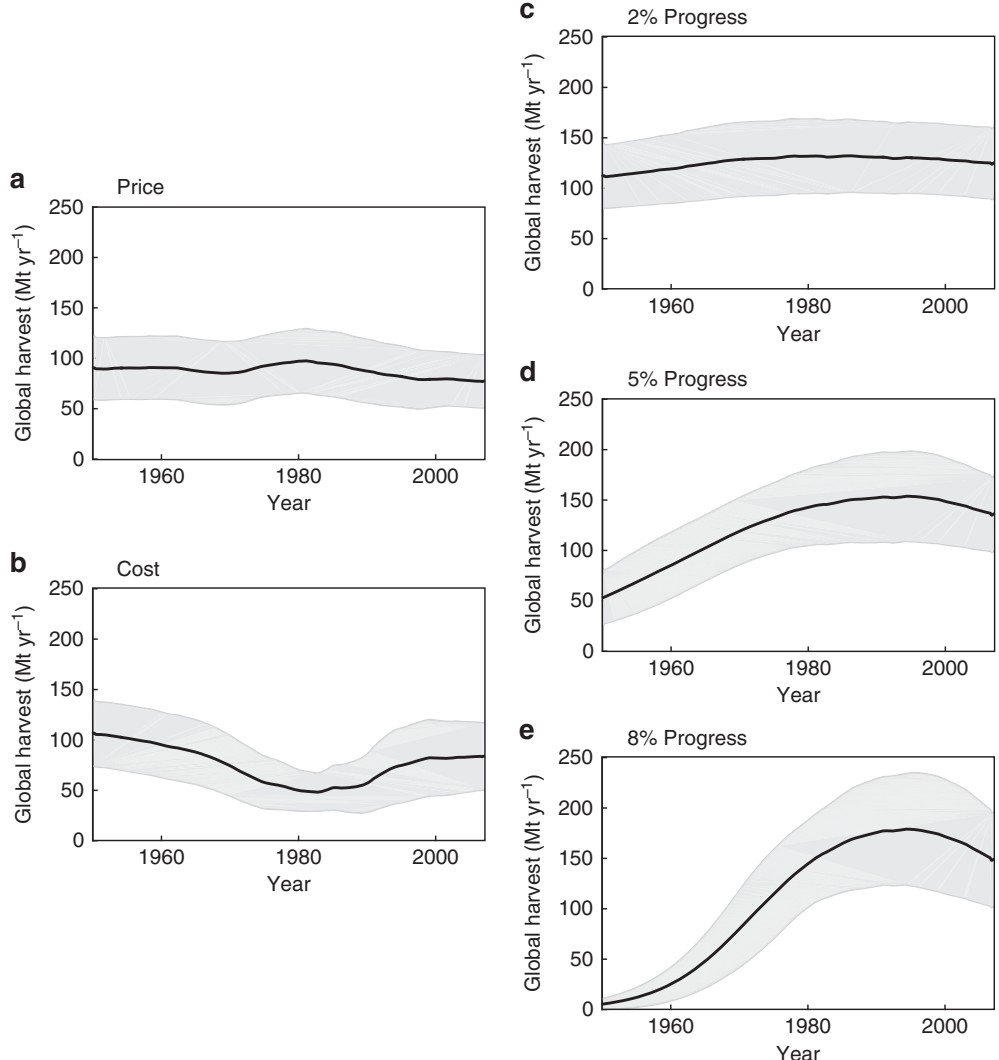

**Figure 2 | Attribution of drivers using model hindcasts of the global fishery.** All simulations assume OA, and each varies only the forcing specified; all other ecosystem and economic dynamics are solved prognostically. The average ex-vessel price of fish (**a**) and cost of fishing per unit effort (**b**) are derived from observations, while technological progress is imposed as a constant rate of catchability increase of (**c**) 2, (**d**) 5 and (**e**) 8% per year.

future trend in fish harvest under open access, but would result in declining, rather than increasing, harvest. The future simulations also assess the relative impacts of climate change on fish harvest, with and without effective management of fisheries, and considering the possibility of increasing market demand.

## Results

**Hindcast simulations.** We carried out a series of experiments in which the model was subjected to temporally varying histories of the three economic forcings: (1) ex-vessel fish price (that is, the price that fishermen receive per mass of catch), which has varied over time in response to market conditions; (2) cost per unit effort, which includes capital, fuel, labour and the impact of most subsidies and (3) technological progress (Supplementary Methods). Our definition of technological progress includes advances in embodied technology, including more efficient boats, more effective fishing gear, sonar and communications equipment and disembodied technology, such as better knowledge of fish behaviour and more efficient fishing practices[24]. The combined effect of embodied and disembodied technology is represented in a simple but inclusive way by a catchability parameter, which was increased at rates of 2, 5 and 8% per year, a

range representative of progress estimated empirically from individual fisheries[7,25–27].

As shown in Fig. 2, the simulated changes in global harvest do not reproduce the observed increasing 20th century trend when forced individually with reconstructed changes in price or cost alone. Given the open-access assumption, this implies that increasing demand for fish was not the main driver of the long-term 20th century increase in fish harvest, despite population growth, although a growing population may have helped to maintain demand in the face of an increasing fish supply. Nor could the reconstructed changes in the cost per unit effort of fishing have been the primary driver. In contrast, the three simulations with technical progress produce global histories of harvest with long-term 20th-century increases, more consistent with the observations. With increasing technology the model also simulates a global peak of harvest, as observed, which arises from the sequential development, overexploitation and/or collapse of fisheries throughout the world, a sequence that has occurred historically in poorly managed fisheries[28,29] (Supplementary Methods).

The technological progress rate of 5% per year would appear most representative as a global, long-term average, given that it

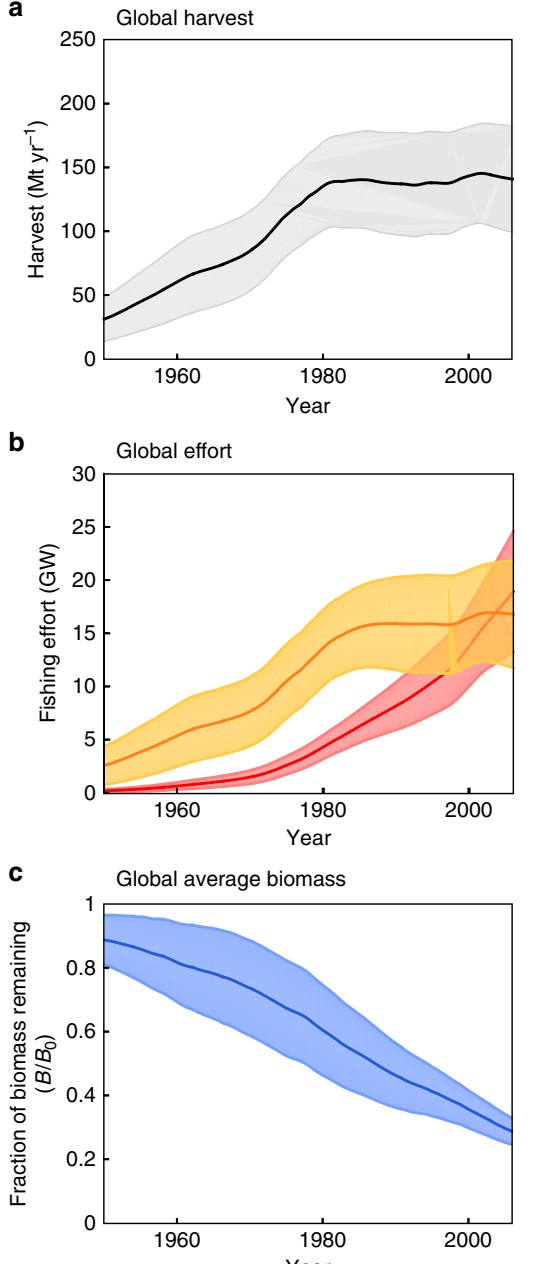

**Figure 3 | Global historical hindcast simulation.** Lines and shaded areas show the model ensemble mean and 1 s.d., respectively, assuming open access (OA, that is, no regulation), 5% per year technological progress, the observed history of ex-vessel price, and historical climate variability as simulated by the IPSL climate model. (**a**) Simulated fish catch per year. (**b**) Nominal effort (orange) and effective effort (red). (**c**) Simulated biomass as a fraction of pristine biomass. All model forcings and parameters other than technology, price and climate are held constant throughout the 56-year simulation.

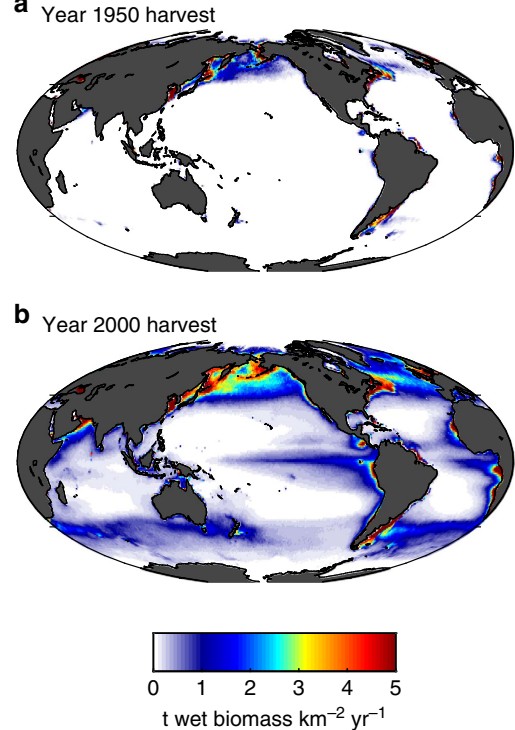

**Figure 4 | Spatial expansion of fisheries in historical hindcast simulation.** Ensemble-average harvest in years 1950 (**a**) and 2000 (**b**). Colour shading shows the harvest in t km$^{-2}$ yr$^{-1}$. For these plots, the ensemble was forced with satellite-based observational estimates of primary productivity and temperature, subjected to globally homogeneous historical price variations and steady 5% per year technological progress. Although idealized, the simulations reproduce key aspects of late 20th century changes, including a shift to lower latitudes and deeper waters.

approximates the observed relative increase of harvest between 1950 and 1996 and is near the midpoint of estimates from individual fisheries[7,27,30]. Given this apparent support for a long-term average technological progress rate of roughly 5% per year, we apply it together with the historical price reconstruction to generate a standard global hindcast, shown in Fig. 3. The agreement of the hindcast simulation with the reconstructed fish harvest, effort and biomass (compare Fig. 3 with Fig. 1) is remarkable, given that these are emergent properties of the model. The effort estimated before 1970 ($\sim$7 GW) is notably higher than that simulated by the model ensemble (1–5 GW), but given difficulties in reconstructing historical global fishing effort[2], these contrasts may not be significant. The estimated rate of biomass decrease is well-reproduced by the model at $\sim$10% of pristine biomass per decade, but with an important difference following 1990 when the estimated rate of biomass loss slowed (Fig. 1c), in contrast with the model. This divergence would be consistent with a progressive improvement in the management of many fisheries over the past three decades, a change that is not captured by the idealized OA simulations.

The model simulations also reproduce the first-order aspects of the 20th-century spatial changes in fish harvest, first depleting the dense biomass of highly productive coastal waters[31], and then moving into deeper waters of the open ocean[32,33] (Fig. 4). The expansion away from the coasts arises in the model as technological progress makes open-ocean waters with low biomass density more profitable, and therefore increasingly subjected to fishing effort. It should be emphasized that this model does not account for higher costs of fishing far from port, nor does it include complex ecosystem interactions, habitat alteration, fisheries management, or spatially variable economic factors, all of which played some role in the history of the global fishery. Nonetheless, it successfully reproduces the first-order features of the historical global trends in harvest, effort, and biomass. We therefore hypothesize that technological progress, at an average rate of $\sim$5% per year, dominated the development of the global wild capture fishery during the 20th century, while other societal, economic and climate forces had secondary impacts.

**Idealized future projections**. We consider an extreme range of possible futures for the global wild-capture fishery by projecting the model forward under idealized economic scenarios, with and without climate change. First, the historical OA hindcast is projected forward to represent an extreme end-member in which management is absent, under two alternative technological progress scenarios, and with either constant or linearly increasing price (Methods). Second, the model is used to estimate the global Maximum Sustainable Yield (MSY), representing the theoretical upper limit that could be approached with perfectly effective and well-informed management aimed at maximizing food production (Supplementary Methods). We emphasize that the future of the global fishery will follow neither the purely OA nor the MSY pathway; rather, these end-members outline the boundaries of what is possible for the global wild fishery in terms of food production. For example, effective regulations aimed at maximizing economic yield would produce less than the MSY harvest, but would generate more profit for fishermen[19]. We simultaneously consider the impacts of climate change by using water temperatures and primary production from the Institut Pierre Simon Laplace (IPSL) Earth System model with the RCP8.5 scenario (high emissions), and compare with a stable pre-industrial climate (Methods, Supplementary Methods).

The MSY is determined exclusively by the ability of the ecosystem to produce harvestable biomass, so that this modelled upper limit of potential harvest depends only on climate. As shown by Fig. 5a, the model ensemble estimates a MSY of $\sim 180 \pm 60$ Mt yr$^{-1}$ under stable preindustrial climate, while the inclusion of historical climate change leads to a decrease of MSY to $160 \pm 50$ Mt yr$^{-1}$ by 2015. The present-day global MSY estimate of 123 Mt yr$^{-1}$ provided by ref. 19 lies within the 1 s.d. range of the model ensemble. Under the rapid continuing climate change of the RCP8.5 scenario, MSY decreases by 20% relative to 2015 by the end of the 21st century, falling below the simulated peak harvest obtained under OA, consistent with the view that climate change will cause a significant drop in potential fish production without reduction of carbon emissions[34].

However, a much more dramatic decline in global harvest occurs over the 21st century in the hypothetical cases for which effective management is absent, as illustrated by the OA scenarios. Given continued 5% per year technological progress and constant price, the peak in global harvest is followed by a long-term decline (Fig. 5a, grey). This reversal in the role of technological progress—from enhancing to reducing harvest—arises because the ability of most ecosystems to produce biomass becomes limited by the spawning stock size under intense rates of harvest, as shown in many scientific surveys of fisheries[35,36], and constrained here through our calibration against observed harvests (Methods, Supplementary Methods). If technological progress is halted, the harvest stabilizes, with only a small downward drift that is due to climate change (Fig. 5a, green). In the OA simulations where future price increases, shown in Fig. 5b, the same general trajectories are maintained, but the decline of harvest is more severe because over-fishing is intensified.

## Discussion

Our hindcast simulations, forced with relatively simple historical scenarios, suggest that a long-term average technological progress rate of roughly 5% per year dominated the 20th century trend in global fish harvests. In reality, technological improvements did not proceed by a steady march synchronized throughout the world, but proceeded at heterogeneous rates among different fisheries according to the development and diffusion of new ideas, and access to the capital required to implement them.

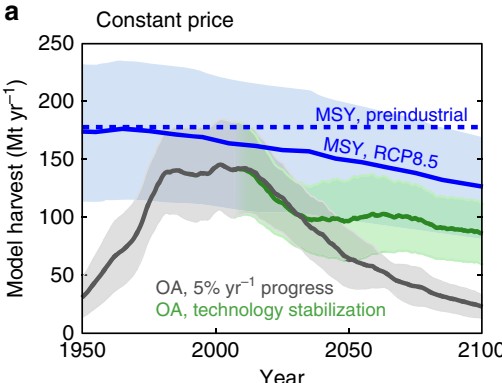

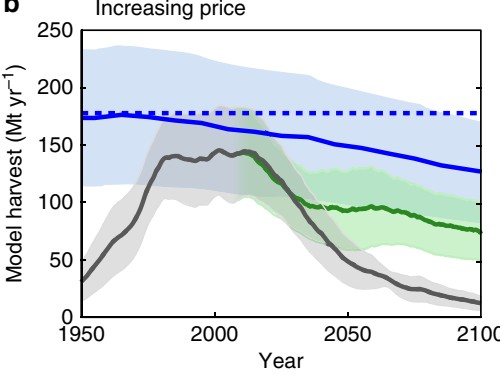

**Figure 5 | Future ensemble projections of global harvest under idealized scenarios.** In each panel, four different idealized scenarios illustrate a range of possible long-term outcomes, reflecting different roles of technological progress, fisheries management, climate change and market conditions. Black lines show OA simulations, representing an absence of management, with steady 5% per year technological progress. Green 'technology stabilization' OA simulations undergo a gradual decrease in technological progress from 5% per year in 2006 to zero by 2036, after which the technology is held constant. The global MSY that could be achieved given perfect management aimed at maximizing harvest is shown in blue. For the OA and MSY RCP8.5 scenarios, the 1 s.d. range of the five ensemble members is shown by shaded envelope. (**a**) Simulations with constant future price, (**b**) simulations with linearly increasing price from $\sim 1\$$ kg$^{-1}$ in 2006 to $\sim 3\$$ kg$^{-1}$ in 2100. All OA simulations, as well as the MSY RCP8.5 simulation, use the IPSL Earth System model climate change projection, while the preindustrial MSY is plotted as a constant value, calculated from the mean of the period 1850–1900 in the IPSL historical simulation.

Nonetheless, given the large number of technological innovations that took place over the 20th century, from monofilament lines, power blocks and new hook designs[37], to sonar, radar and satellite communications, it is plausible that the overall aggregate progress—integrated among industrial fisheries of the world—may have been relatively smooth.

The impact of technological progress is achieved in the model by two interdependent pathways: fish become easier to catch for a given nominal effort and fish biomass, increasing effective effort, which in turn makes fishing more profitable, thereby increasing the nominal effort as long as the ecosystem is capable of sustaining the harvest. The fact that, over a 50-year period, a 5% per year rate of technological progress would have increased catchability by an order of magnitude explains its dominance over the relatively small historical fluctuations in the ex-vessel price and the cost of fishing.

Our idealized future simulations suggest that, over the 21st century, continuing technological improvements would cause

global harvest to decrease under OA. This reversal of the 20th century trend would be exacerbated by any market dynamics that raise prices. These simulated future decreases, which lead to a growing divergence between OA and MSY over time (Fig. 5), suggest that the global impact of effective regulation will become more important in future as technology improves: whereas in 2015, the simulated MSY harvest is only ~15% greater and ~$20 billion per year more profitable than the global OA harvest, in 2100 (given 5% per year progress and increasing prices) the MSY harvest is more than ten-fold greater and hundreds of billions per year more profitable than the OA harvest. Although this may be of only theoretical interest for fisheries where effective regulatory regimes are well-established, it could have serious food security implications for the numerous fisheries in which regulations remain ineffective. The model suggests that, if perfectly managed, the global MSY would be on the order of 25% greater than the OA historical peak under preindustrial climate, and that even with the rapid climate change of the RCP8.5 scenario, the MSY would remain greater than the OA historical peak throughout most of the 21st century. Thus, our model suggests that the global wild capture fishery could continue to provide food, throughout the 21st century, at the same rate (or better) as at the end of the 20th century, but only if effective management can be extended to the fisheries in which it remains weak or absent.

We note that if new fisheries develop for previously unexploited species (such as mesopelagic fish), not included in our model, this could potentially slow the reduction of harvest under OA and raise global MSY. At the same time, our model is generally optimistic, in that it does not resolve habitat destruction by intensive trawling[38], the takeover of ecosystems by non-commercial species[39], trophic cascades[40], alternative stable states[41], ocean acidification and deoxygenation, or the development of overcapacity due to subsidies[6], all of which would benefit from further study. The urgent need for improved fisheries management at the global scale is widely recognized, given the existing level of fishing effort[19] and growing pressure from climate change[42,43]. Our results amplify these concerns by showing that, whereas progress in fishing technology contributed to massive gains in fish harvest during the 20th century at the global scale, its influence on catchability would be expected to only reduce harvest in the 21st century unless met with a global expansion of effective fisheries management.

The model presented here takes a highly simplified approach to the wild capture fishery, as required to remain tractable at the global scale and over long timescales. But despite its simplicity, the comprehensive treatment of critical elements, including spatially and temporally resolved interactions of humans with the ecosystem, allow it to identify the relative importance of general mechanisms that are difficult to discern among the multiplicity of factors typically considered on shorter timescales. Similar approaches may help to better understand the long-term dynamics of other coupled human-Earth systems.

## Methods

**Model overview.** At the foundation of the model is energy production by phytoplankton, which is transferred to multiple fish size-spectra according to an efficiency that depends on gross ecosystem metabolic rates[16,44,45]. The fish spectra are defined as representing the entire range of finfish and invertebrates with individual mass greater than 10 g, commercially harvested before 2006. The spectra resolve basic life-history characteristics, including temperature-dependent growth, mortality and density-dependent recruitment. By including all commercial species within the spectra, we implicitly account for species range shifts[46,47], evolution[48], and changes in targeted species, all of which have been significant historically and could play important roles in the future.

An economic model, coupled directly to the fish spectra at the grid scale, predicts fishing effort as a function of local profit, assuming open access to the available fish, following the classic work of ref. 49. Fishing effort in each grid cell

changes over time according to the difference between local revenues and costs, at a rate determined by the fleet adjustment timescale. The effectiveness of fishing effort at catching the locally available biomass is determined by a catchability parameter that encapsulates both embodied and disembodied technology. Although fisheries are not strictly open access, and approximately one third of fisheries currently has some form of reasonably effective management[19], the absence of effective management from most of the world's fisheries throughout the 20th century makes it a good approximation at the global scale during the historical period[21,22] as well as an illustrative example to consider the importance of management in future. The approximate global Maximum Sustainable Yield (MSY) is calculated by conducting transient simulations in which catchability is increased slowly, and summing the maximum harvest obtained in each grid cell. A more complete description of the model is given in 'Model description' in Supplementary Methods.

**Parameter selection.** We calibrate the most important 13 model parameters through a Monte Carlo method and comparison with observationally estimated fish catches and stock assessments at the LME scale ('Parameter optimization' in Supplementary Methods). Because the LMEs span a very broad range of temperature and primary production, this strategy ensures a robust calibration to both these important variables. For the simulations shown here, we use an ensemble of five optimized models that span the uncertain parameter space as widely as possible, while producing a reasonably accurate simulation.

**Model forcing and simulations.** The model is forced with ocean temperature and net primary production, either from satellite-based observational estimates, or from the Institut Pierre Simon Laplace (IPSL) global climate model[50], using the historical hindcast followed by the Representative Concentration Pathway (RCP) 8.5 scenario. The factors at play during the historical transient, according to the model framework, include the increase of catchability due to technology[24,26], and an observational estimate of the global average ex-vessel fish price[51], which has varied remarkably little ('Ex-vessel Price variations' in the Supplementary Methods). Given that there is no currently available estimate of historical variations in cost per unit effort, we roughly approximate its potential role using its relationship with observed price, effort and harvest under open-access, and making a steady-state assumption ('Historical global average cost per unit effort' in Supplementary Methods).

For the future projections, the OA model is integrated under four idealized scenarios, two of which assume a continuation of the same 5% per year exponential increase in fishing technology, while the other two simulate a gradual decrease in the rate of progress after 2006, reaching stabilized-technology after 2036. Given that technological progress is clearly continuing through advances such as improved fish tracking, the deployment of fish aggregating devices, and the increasing mechanization of fleets in developing nations, the technology-stabilization scenarios are included as baselines from which to estimate the long-term importance of future technological progress, rather than representing realistic future outcomes. For each of the technological scenarios, we address uncertainty in future market conditions by changing the ex-vessel price. Because the price of fish depends on the demand for fish products, which is difficult to predict given its dependence on societal preferences, available substitutes, and distribution networks[52], we apply two end-member scenarios for ex-vessel price: linearly increasing, and constant. These represent the outcomes that might come from a larger population and/or a greater preference for seafood (increasing price), or a ready availability of substitutes from terrestrial food production (constant price), and bracket a wide range of intermediate possibilities.

**Code availability.** The model code was written in MATLAB version R2012a. The zero-dimensional version of the model (that is, for a single patch of ocean), which includes the model run script, required functions, and forcing data, is available for download at http://dx.doi.org/10.5281/zenodo.27700 (ref. 53).

**Data availability.** The model output and observational data are available from the authors on request.

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

## Acknowledgements

We thank the Canadian Foundation for Innovation (CFI) for providing computing infrastructure, and the Marine Environmental Prediction and Response (MEOPAR) Network Centre of Excellence and Canadian Foundation for Advanced Research (CIFAR) for funding support. EDG acknowledges financial support from the Spanish Ministry of Economy and Competitiveness, through the María de Maeztu Programme for Centres/Units of Excellence in R&D (MDM-2015-0552). This project has received funding from the European Research Council (ERC) under the European Union's Horizon 2020 research and innovation programme (grant agreement No 682602, BIGSEA). D.B. acknowledges financial support from a Faculty Research Grant from University of California Los Angeles.

## Author contributions

E.D.G., D.A.C. and D.B. designed the study. D.A.C. and D.B. wrote the model code. D.A.C. and D.B. performed the model simulations. E.D.G., D.A.C. and D.B. contributed to the analysis. E.D.G. wrote the paper with contributions from D.A.C. and D.B.

## Additional information

**Competing interests:** The authors declare no competing financial interests.

