## [Peer Review File · Nature Communications]

Reviewers' comments:

Reviewer #1 (Remarks to the Author):

In this manuscript, the authors simulate the development of world fisheries using a bioeconomic marine trophic size-spectrum model. The model has two components, namely a very detailed dynamic spatial biological size-based part and a very simple economic part, which is combined using the standard linear harvest function in fisheries. This is done at a grid cell level. Fish prices and unit fishing cost are the same across grid cell. There is no regulatory framework (i.e. open access is assumed) imposed.

I'm impressed by the biological part of the model being both size-based and spatial, but also less impressed by the economic part. I will come back to this.

The theory from fisheries economics tells us and predicts in general that under technological progress, the open access solution will be a function of time and over time lead to lower and lower fish stock biomass. The rate of decline in stock biomass and approach path depends on all the bioeconomic parameters including the rate of technological development.

This result is what the empirical model used in the paper confirms. So as such, there is no new insight obtained by the authors. The new is the advanced biological modeling approach at the global level. The authors calibrate the model to the real development in the world's fisheries in terms of harvest and stock size using the rate of technological progress as the calibration variable. They find that a rate at around 5% gives to best fit. It is important to verify theory empirically, but in this case then the authors would have to estimate the rate of technological change and then use it as an input parameter in the model. The point is that their empirical model is a simulation model and cannot be used to verify the prediction of the theory. But they can use the model as they do to make forecast given different parameter values on prices, cost and rate of technological change under open access.

However, the model cannot explain the stabilization of the fish stock biomass (figure 1e) that has taken place since 2000, because the model aggregate biomass (figure 1f) is declining. I suspect that it is because many fisheries are now regulated more effective and efficient. Since the biological part is spatial based, it should be possible to impose different management systems. I suggest keeping international waters as open-access and a significantly share of national waters regulated more efficient, at least in the developed countries. In my opinion, this will also give a more realistic description of the impact of technological progress, because efficient regulations can, if designed properly, transform technological progress to an economic gain. A positive side-effect to the story of the paper is that it can demonstrate the main point for the paper, mainly that well-designed fishery management is needed.

More detailed comments:

- New title is needed. In my opinion the title is for a newspaper. What threatens fish stocks are the combination of technological progress and in-effective and in-efficient regulation.
- The statement "This trend in fish harvest reflects an interaction of global human and

ecological forces that have not previously been considered together within a unified quantitative framework" in the abstract and introduction need to be rephrased, because e.g. the World Bank and the other have published work based on an unified quantitative framework (Willman et.al. 2009 and Sumaila et.al. 2010), but of course with a different approach and purpose. So the statement has to be qualified.

- I suggest to delete "while market forces, population pressure and climate change are likely to make the situation worse" from the sentences "The model shows that further technological progress, which is typically ignored in fisheries economics, can only decrease the wild capture harvest in future unless met with effective regulation, while market forces, population pressure and climate change are likely to make the situation worse", because these drivers are not the main focus of the paper and a little unclear. Why should market forces by it-self make the situation worsen?

- "Because the price of fish depends on a multiplicity of unresolved societal processes" is difficult to understand. In fish markets the price of fish is determined by demand and supply and hence by economic incentives. Please explain.

- In figure 3 there are simulations of different future technological progress and price developments. As I understand the economic model, then technological progress and price increase have the same impact in the model. So, an increase in prices from 1\$ to 3\$ from 2006 to 2010 is the same as an annual rate of technological progress at 1.2% (my calculation to annual rate), figure 3b.

- In SI: ""Price" is that paid to the fishermen at the point of landing (exvessel), and is also spatially uniform, though a temporally-variable price is imposed". I understand that a size-based model is used leading to three aggregate size spectra and the question is whether the assumption of one single constant price across size spectra makes sense? In other words could the price be depending on size?

- Costs are assumed constant across grid cells. I believe that the model could gain from letting some of the variable costs depending on the distance from port.

References:

Sumaila, U. R., Khan, A. S., Dyck, A. J., Watson, R., Munro, G., Tydemers, P., & Pauly, D. (2010). A bottom-up re-estimation of global fisheries subsidies. *Journal of Bioeconomics*, 12(3), 201-225.

Willman, R., Kelleher, K., Arnason, R., & Franz, N. (2009). The sunken billions: the economic justification for fisheries reform. IBRD/FAO.

Reviewer #2 (Remarks to the Author):

The most striking feature of trends in world fish stock abundance and effort is differences between regions as seen in Worm et al. 2009. Costello et al 2012 showed striking differences between stocks that were assessed by scientific agencies, and stocks that were not.

Fish stocks are increasing in abundance throughout most of the developed world as a result of fisheries regulation. Fishing effort has declined dramatically in the N. Atlantic especially,

but also in many other areas, and catches have declined in these regions even though abundance is increasing.

This model does not include fisheries regulation, which now is in place (with some significant failures) in roughly 40% of global fish production. The model does not distinguish between regions where abundance is increasing, and regions where it is declining.

The idea that fisheries can be represented by a global fishing effort is untenable ... fishing effort as measured by fishing mortality rate is declining in most areas where we have good data.

The failure to incorporate fisheries regulation and to show that their model can replicate the major differences in trends in stock abundance and effort between major regions of the world invalidate any conclusions from the analysis.

Worm, B. et al. Rebuilding Global Fisheries. *Science* 325, 578-585 (2009).

Costello, C. et al. Status and Solutions for the World's Unassessed Fisheries. *Science* 338, 517-520 (2012).

Reviewer #3 (Remarks to the Author):

Review of: Technological progress threatens the future of wild fish harvest by Galbraith, Carozza & Bianchi

Main comments:

This paper is an exciting and novel contribution to the marine fisheries literature. It is the first global application of a dynamic hindcast model that couples human ecosystem interactions to economic drivers of fisheries by integrating dynamics for physical, biogeochemical, and size-structured food web processes with technological development of fisheries. The global model reproduces empirical -based time series and important shows that a 5% per year increase in technology is needed to reproduce past change (and it does not emerge from changes in price or nominal effort). Under climate projection experiments, the effects of continued changes in technology are shown to lead to potential future collapse. Technological change was shown to have much greater consequences than changes price or even climate change scenarios when considered separately. Although the model assumes open access fisheries rather than management, the results highlight the importance of management and in particular the need for appropriate control of changes in technology to catch fish.

Overall the paper is well written and clear but there are few places where the methodology needs to be clearer in order to be transparent. Although I appreciate the authors have provided a great amount of detail in related documentation provided, it would be nice to

have enough of the detailed self-contained even if in the SI. I have also suggested some additional papers be cited, particularly recent climate change fisheries projections and bio-economic modelling work, that I felt were missing (see specific comments below). Having said all that, I think addressing these issues is fairly straightforward and if published, I think this paper is likely to become one of the top fisheries papers of this year. It was a pleasure to review this paper.

Specific Comments

Page 3 - line 17 - "The model is unique in that it comprehensively considers both human and natural components of the system, using relatively simple but well-founded principles". This isn't quite true, that is, in terms of being the only model to represent both human and natural components. Acknowledgement of the recent literature and use of similar methods for addressing related questions, seems to be missing in the paper, but would actually help to justify the novelty of your approach even further. For example, in previous work 'time-slice' hind-casts and projections have been used, with simple size-based models validated using time-averaged data for Large Marine Ecosystems and EEZs, rather than attempting to hindcast time series of global catches (Barange et al. 2014 *Nat. Clim. Change*, 4(3): 211-216. doi:10.1038/NCLIMATE2119; Merino et al. 2012 *Global Environmental Change*, 22 (4), 795-806). The linkage to the human component in those studies is also included but is one-way, rather than explicitly being coupled to the natural system enabling the feedbacks between them to be captured - eg. a major advancement in your study. Furthermore, yours is novel because it is the first global application of its kind to predict long-term temporal changes in global fisheries, that successfully reconstructs observed past changes through time, through calibration of the model with data (before doing the projections)) and importantly through explicit consideration of the effects of technological change.

In terms of recent work on the importance of changes in technological efficiency, this paper might also be useful: <http://icesjms.oxfordjournals.org/content/73/4/1226.full>

Page 3 - Figure 1 caption: the model ensemble is mentioned here but it isn't until you look at the SI that you realise this is a single model ensemble (e.g. models from different parameter sets rather than for example different model structures or from different climate model inputs). This could be more clearly stated by adding a sentence, perhaps after at the top of page 4, after the calibration text. Just a minor typo, but the reference numbers aren't in subscript in the caption.

Page 4 - Line 1 - it is not clear what kind of approach for predicting changes in growth is used? Looking at the SI and the additional model documentation in a different paper, the main equation is McKendrick von Foerster but it differs from the trait-based and size-structured food web model of Hartvig et al. 2011 (*J. Theor. Biol.* 272(1): 113-122. doi:10.1016/j.jtbi.2010.12.006) which has food dependent growth etc. More detail on the approach should be provided, even if briefly so that readers understand the model when reading this paper. All of the model equations and parameters (with definitions) should ideally be included in the SI (perhaps these could be confined to tables to prevent using too

much space) with enough info in the main text to get the jist. I commend the authors for also making everything available on GitHub, but some readers won't go to the extra effort to look at code.

Page 4- line 3, same comment as above... more details on how fishing effort was modelled as this isn't clear when consider the main text + SI on their own. providing the equations, including information on how q and E were forced dynamically need to be clearer. Also in the SI peraps the more descriptive text would be better placed

Page 4 - also line 3 - re: "poorly constrained parameters" Need to state which parameters were estimated, which ones were not, what are the different parameters. Again model details as suggested above would solve this problem

Page 4 - perhaps replace "robust optimisation" with optimisation based on a set of model selection criteria (is it history matching?) ... otherwise sounds like it's least squares or Bayesian , but it's not clear

Page 4, line 17 - The model is size-structured but I couldn't find any mention of the size selectivity of the gear, which has been explored a lot with size spectrum models (Scott et al. 2014, *Methods Ecol. Evol.* 5(10): 1121-1125. doi:10.1111/2041-210X.12256, Jacobsen, N.S., Gislason, H., and Andersen, K.H. 2014. *Proc. R. Soc. B Biol. Sci.* 281(1775): 20132701. doi:10.1098/rspb.2013.2701) including a recent paper that also considers bioeconomic drivers (Andersen, K.H., Brander, K., and Ravn-Jonsen, L. 2015. *Ecol. Appl.* 25: 1390-1396. doi: 10.1890/14-1209.1). These are all covered in a recent review by Andersen et al. 2016 (*CJFAS*, 73(4): 575-588, 10.1139/cjfas-2015-0230). Please can this be clarified somewhere either in the main text or the SI (in general I found the fishing component not that clear but it needs to be as it is crucial to the study).

Page 5 line 10, "Given these results, we apply the hindcast..." In this pgh it became unclear whether the pgh above was also the global model hindcast. It seems that it was more about calibration not the complete reconstruction. Perhaps it is better to move this paragraph up, as it is a big main result and cites Figure 1, then report the details of what was needed to achieve it through the calibration (e.g. swap order of the paragraphs?).

Page 6, line 1 - again interested to know under what types of size selectivity (as size of fish also matters for price) the study assumes, and whether or not it matters...

Page 8, line 3. " The small fish biomass" Up to here and in the SI as far as I can tell the prediction is total biomass across a wide range of sizes (small, medium and large fish) so this sentence is a bit confusing...did you mean to refer to small fish here?

Page 9, line 2 - in terms of highlight outstanding issues for future it would be nice to add here: or feedbacks due to cascading effects of fisheries (Andersen et al 2010), alternative stable states or regime shifts (De Roos et al 2006, *Proc Roy Soc B*); Horan et al 2011 *PNAS*)

SI comments:

Page 1-2 -Model description. This is not really enough detail for the paper to be stand-alone. I know it's a lot of detail but even just including a table of equations and parameters would help a huge amount and will mean more people will appreciate what you have done (many won't go to the extra effort of checking other papers or code repositories - although it is excellent that you have done that!). Since it is the important highlight of your method a much clearer description of how time varying effort and catchability were linked to the natural ecosystem is needed. Some of the details starting on page 5 (pgh 2) would fit nicely at the end of the model description? This would make the historical forcing section, on page 5 much easier to interpret, as it is not fully clear to me how this was implemented in the current text. Overall it would be very helpful if the model equations, parameter definitions and values (highlighting the ones being calibrated and estimated values) were all provided.

Page 4 - "continual transient increases" - what does this mean? Is it a linear increase, exponential increase, extrapolation form an existing time series? More detail would help here.

Page 4 - How good are the model predictions? It was unclear what you meant by saying the ability to predict the peak catches (timing or magnitude for example). Perhaps perhaps this is a wider issue of not being able to assess how good the model performs at the LME scale. Maybe you could include a plot or table showing the differences between the observed and predicted values for each LME or at least reports some aspects of model skill that pertain to the criteria you listed under "Parameter optimization".

We thank the three reviewers for their comments on our manuscript. We have made a very thorough revision that incorporates all of their constructive suggestions, and which we feel leads to a much improved article.

Most importantly, we now include an alternative economic solution to the model, giving Maximum Sustainable Yield (MSY), in addition to the Open Access (OA) dynamic. Calculating the MSY solution required the development of a new methodology, and involved running a large number of new simulations. This major addition is discussed below, in addition to numerous other changes made to the manuscript.

As well as the changes discussed here, we have made many small improvements to the manuscript, in the interests of clarity, nuance and accuracy. We have also reduced the length of the abstract in view of the Nature Communications limits. These changes can be seen in the tracked-changes versions, that follow at the end of this response letter.

Reviewers' comments:

Reviewer #1 (Remarks to the Author):

In this manuscript, the authors simulate the development of world fisheries using a bioeconomic marine trophic size-spectrum model. The model has two components, namely a very detailed dynamic spatial biological size-based part and a very simple economic part, which is combined using the standard linear harvest function in fisheries. This is done at a grid cell level. Fish prices and unit fishing cost are the same across grid cell. There is no regulatory framework (i.e. open access is assumed) imposed.

I'm impressed by the biological part of the model being both size-based and spatial, but also less impressed by the economic part. I will come back to this.

The theory from fisheries economics tells us and predicts in general that under technological progress, the open access solution will be a function of time and over time lead to lower and lower fish stock biomass. The rate of decline in stock biomass and approach path depends on all the bioeconomic parameters including the rate of technological development.

This result is what the empirical model used in the paper confirms. So as such, there is no new insight obtained by the authors. The new is the advanced biological modeling approach at the global level. The authors calibrate the model to the real development in the world's fisheries in terms of harvest and stock size using the rate of technological progress as the calibration variable. They find that a rate at around 5% gives to best fit. It is important to verify theory empirically, but in this case then the authors would have to estimate the rate of technological change and then use it as an input parameter in the model. The point is that their empirical model is a simulation model and cannot be used to verify the prediction of the theory. But they can use the model as they do to make forecast given different parameter values on prices, cost and rate of technological change under open access.

We generally agree with these comments. Essentially, the estimation of the rate of technological change has been provided by prior workers, and found to cluster around 4 or 5% (with outliers at 2 and 8%), as cited in the text. As such, the fact that the 5% progress rate fits best with the history of harvest is reassuring, but not a proof in itself. However, we feel that our calculation showing that this rate of technological change, alone, is capable of explaining most of the historical development of the global fishery, including the spatial evolution, is an important one to make - and one which has not been clearly shown by prior workers. Nonetheless, we agree that the projections we present are the most novel aspect of our work.

However, the model cannot explain the stabilization of the fish stock biomass (figure 1e) that has taken place since 2000, because the model aggregate biomass (figure 1f) is declining. I suspect that it is because many fisheries are now regulated more effectively and efficiently.

We agree that this is an important factor, and although there is possibility for bias, could be the explanation for the stabilization. We have therefore rephrased this:

Original phrase: “The loss of biomass appears to have slowed in the most recent years of the observational estimate, but this may reflect a bias of the available stock assessment data to relatively stable, well-managed fisheries^(Costello, Lynham et al. 2010).”

New phrase: “We note that the loss of biomass has slowed in the most recent years of the stock assessment composite (Figure 1e), which likely reflects an improvement in the management of some fisheries, a change that is not captured by the OA model.”

Since the biological part is spatial based, it should be possible to impose different management systems. I suggest keeping international waters as open-access and a significantly share of national waters regulated more efficiently, at least in the developed countries. In my opinion, this will also give a more realistic description of the impact of technological progress, because efficient regulations can, if designed properly, transform technological progress to an economic gain. A positive side-effect to the story of the paper is that it can demonstrate the main point of the paper, mainly that well-designed fishery management is needed.

We very much agree with the intention expressed here. However, we feel that the suggested method - although a good idea - would not be appropriate in our modeling framework. Our approach differs from prior works on the global fishery because it focuses on emergent properties of fundamental processes, rather than trying to impose ‘realistic’ answers by prescribing them a priori. As emphasized by Reviewer #3, this is an important novelty of our work. To take an analogy from climate science, we are attempting to build something like a General Circulation Model, rather than a statistical model - we therefore want to avoid imposing arbitrary regional rules. Our motivation for

this approach is that models built on fundamental processes have greater predictive power.

Nonetheless, we recognize that including an illustration of the benefits to be gained from well-designed fishery management is an excellent suggestion. We have therefore conducted new simulations that calculate a Maximum Sustainable Yield (MSY). To do so, we developed a novel method that diagnoses the MSY in each grid cell, as a function of climate state, for each ensemble member. These were then aggregated to provide global MSY estimates, shown on the revised Figure 3. We feel that this is a major improvement to the paper, and thank the Reviewer for the encouragement to pursue this important aspect.

More detailed comments:

- New title is needed. In my opinion the title is for a newspaper. What threatens fish stocks are the combination of technological progress and in-effective and in-efficient regulation.

We see the reviewer's point, and have changed the title accordingly.

- The statement "This trend in fish harvest reflects an interaction of global human and ecological forces that have not previously been considered together within a unified quantitative framework" in the abstract and introduction need to be rephrased, because e.g. the World Bank and the other have published work based on an unified quantitative framework (Willman et.al. 2009 and Sumaila et.al. 2010), but of course with a different approach and purpose. So the statement has to be qualified.

This is a very good point. The original statement has been removed entirely, and this distinction has been made at other relevant points in the main text.

- I suggest to delete "while market forces, population pressure and climate change are likely to make the situation worse" from the sentences "The model shows that further technological progress, which is typically ignored in fisheries economics, can only decrease the wild capture harvest in future unless met with effective regulation, while market forces, population pressure and climate change are likely to make the situation worse", because these drivers are not the main focus of the paper and a little unclear. Why should market forces by it-self make the situation worsen?

We can see why this was not clear, as originally phrased. The intention was to refer to the fact that increasing prices lead to less fish, so that - given the assumption that the future will include greater demand, and therefore higher prices - market forces will make the situation worse. It has now been removed from the abstract.

- "Because the price of fish depends on a multiplicity of unresolved societal processes" is difficult to understand. In fish markets the price of fish is determined by demand and supply and hence by economic incentives. Please explain.

We had intended to refer to the fact that the prediction of demand is extremely difficult, given that it depends on societal preferences, available substitutions, and transportation/processing networks, which are challenging to predict on a multi-year timeframe, let alone a centennial timeframe. This has been rephrased to better reflect the intention, as:

“Because the price of fish depends on the demand for fish products, which is difficult to predict given its dependence on societal preferences, available substitutes, and distribution networks...”

- In figure 3 there are simulations of different future technological progress and price developments. As I understand the economic model, then technological progress and price increase have the same impact in the model. So, an increase in prices from 1\$ to 3\$ from 2006 to 2010 is the same as an annual rate of technological progress at 1.2% (my calculation to annual rate), figure 3b.

The impact of progress and price are different, in that price affects only the nominal effort, whereas progress affects both the nominal effort and the effective effort. However it is true that the imposed price change is similar to a relatively slow technological progress rate of 1.2%. We realize that, given our lack of predictive power for future price changes (which could potentially be more extreme than the three-fold increase simulated) we should not unequivocally state that technology is more important. The relevant passages have been rephrased accordingly.

- In SI: "Price" is that paid to the fishermen at the point of landing (exvessel), and is also spatially uniform, though a temporally-variable price is imposed". I understand that a size-based model is used leading to three aggregate size spectra and the question is whether the assumption of one single constant price across size spectra makes sense? In other words could the price be depending on size?

Indeed, our original intention was to include size-dependent prices when we started building the model. However, although size can have a clear impact on price among specific types of fish, after doing a detailed survey of historical prices, we failed to find a simple, robust relationship between asymptotic size and price that held across all commercial fish over time. For example, cod and tuna both have large asymptotic size, but cod has tended to be significantly less valuable per kg. On the other end of the spectrum, shrimp have small asymptotic sizes but are extremely valuable per kg. These prices reflect societal preference related to factors such as taste, appearance, convenience or cultural value, and are therefore difficult to predict. Yet, the inflation-adjusted global average price of all fish is remarkably constant over time, consistent with the availability of substitutes, and a distribution of societal preferences across the available fish types. We have added text to the SI in order to briefly discuss this.

- Costs are assumed constant across grid cells. I believe that the model could gain from letting some of the variable costs depending on the distance from port.

We agree that this is a very interesting dynamic, and have made some initial tests using distance maps to port. However, including this dynamic would cause the model to be more complex, and introduces some arbitrariness in terms of port locations, the assumed duration of fishing trips, and the costs of fuel vs. capital, maintenance and labour. At the same time, including the distance to port as a fraction of the cost per unit effort would not have a significant impact on the main results under discussion here. We therefore prefer to leave this topic for future work where it can be developed properly.

References:

Sumaila, U. R., Khan, A. S., Dyck, A. J., Watson, R., Munro, G., Tydemers, P., & Pauly, D. (2010). A bottom-up re-estimation of global fisheries subsidies. *Journal of Bioeconomics*, 12(3), 201-225.

Willman, R., Kelleher, K., Arnason, R., & Franz, N. (2009). The sunken billions: the economic justification for fisheries reform. IBRD/FAO.

Reviewer #2 (Remarks to the Author):

The most striking feature of trends in world fish stock abundance and effort is differences between regions as seen in Worm et al. 2009.

Costello et al 2012 showed striking differences between stocks that were assessed by scientific agencies, and stocks that were not.

Fish stocks are increasing in abundance throughout most of the developed world as a result of fisheries regulation. Fishing effort has declined dramatically in the N. Atlantic especially, but also in many other areas, and catches have declined in these regions even though abundance is increasing.

This model does not include fisheries regulation, which now is in place (with some significant failures) in roughly 40% of global fish production. The model does not distinguish between regions where abundance is increasing, and regions where it is declining.

We entirely agree with the reviewer's point that a fraction of the world's fisheries now have management that is effective to some degree, and certainly did not mean to imply otherwise. We note that a new paper by Costello et al. (PNAS, 2016) estimates that 32% of fisheries are currently in good biological condition. However this still leaves on the order of 2/3 of the global fisheries without poor management, and therefore somewhere between the MSY and the open access solution.

We would emphasize that our use of an open-access model is not meant to be an accurate representation of the present, or an accurate prediction of the future. Rather, it is more reflective of the historical development of the global fishery during the 20th century, when management was rare and largely ineffective. It is also meant to show the

interaction between technological progress, climate change, and open access, in order to illustrate the importance of management. As such, it represents a new line of support for the importance of regulation, to which the reviewer refers. The reviewer's comments clearly show that these nuances were not clear in the original manuscript, and we have modified the manuscript in many places in order to better convey this intention.

In addition, we address the reviewer's concern regarding the importance of fisheries regulation through our inclusion of the new simulations estimating global MSY, as discussed in the response to Reviewer 1. The MSY calculation can be interpreted as a target that could be achieved through effective governance at the global scale, aimed at maximizing food production.

The idea that fisheries can be represented by a global fishing effort is untenable ... fishing effort as measured by fishing mortality rate is declining in most areas where we have good data.

First, this concern reflects a misunderstanding of our model. We entirely agree that using a single global fishing effort would be an unacceptable simplification, and for this reason the model calculates an independent, dynamic nominal effort in each grid cell. The effective effort is given by the product of the dynamic nominal effort and the catchability factor.

Second, as mentioned above, the OA solution is not meant to be a prediction: it provides an illustrative scenario to quantify the importance of regulation, and showing that this importance will only increase in the future as technology continues to progress. We feel this point is now made much more clearly, with the inclusion of the MSY calculation.

The failure to incorporate fisheries regulation and to show that their model can replicate the major differences in trends in stock abundance and effort between major regions of the world invalidate any conclusions from the analysis.

Although we agree that it would be useful to pursue more intricate models that explore regional differences in management, we strongly disagree that this is a necessary consideration to support our conclusions. As pointed out in the comments to Reviewer #1, the strength of our approach lies in its basis on fundamental principles, as a means to identify emergent behaviour. The open access solution is highly relevant for the poorly-managed fisheries that unfortunately persist in much of the world. In addition, there is no guarantee that fisheries that presently have successful management will continue to be successfully managed in future, given the pressure of increasing technology - this will presumably require continued vigilance. Our results provide dramatic support for the need for this vigilance, by quantifying the huge gains to be made with effective management.

Worm, B. et al. Rebuilding Global Fisheries. Science 325, 578-585 (2009).

Costello, C. et al. Status and Solutions for the World's Unassessed Fisheries. Science 338, 517-520 (2012).

Reviewer #3 (Remarks to the Author):

Review of: Technological progress threatens the future of wild fish harvest by Galbraith, Carozza & Bianchi

Main comments:

This paper is an exciting and novel contribution to the marine fisheries literature. It is the first global application of a dynamic hindcast model that couples human ecosystem interactions to economic drivers of fisheries by integrating dynamics for physical, biogeochemical, and size-structured food web processes with technological development of fisheries. The global model reproduces empirical -based time series and important shows that a 5% per year increase in technology is needed to reproduce past change (and it does not emerge from changes in price or nominal effort). Under climate projection experiments, the effects of continued changes in technology are shown to lead to potential future collapse. Technological change was shown to have much greater consequences than changes price or even climate change scenarios when considered separately. Although the model assumes open access fisheries rather than management, the results highlight the importance of management and in particular the need for appropriate control of changes in technology to catch fish.

Overall the paper is well written and clear but there are few places where the methodology needs to be clearer in order to be transparent. Although I appreciate the authors have provided a great amount of detail in related documentation provided, it would be nice to have enough of the detailed self-contained even if in the SI. I have also suggested some additional papers be cited, particularly recent climate change fisheries projections and bio-economic modelling work, that I felt were missing (see specific comments below). Having said all that, I think addressing these issues is fairly straightforward and if published, I think this paper is likely to become one of the top fisheries papers of this year. It was a pleasure to review this paper.

We thank Reviewer #3 for the enthusiastic support of the paper. We recognize the value of including more methodological details, and have greatly expanded the description of the model and the Monte Carlo optimization procedure in the revised supplement, as suggested.

Specific Comments

Page 3 - line 17 - "The model is unique in that it comprehensively considers both human and natural components of the system, using relatively simple but well-founded principles". This isn't quite true, that is, in terms of being the only model to represent both human and natural components. Acknowledgement of the recent literature and use of similar methods for addressing related questions, seems to be missing in the paper, but would actually help to justify the novelty of your approach even further. For example, in previous work 'time-slice' hind-casts and projections have been used, with simple size-based models validated using time-averaged data for Large Marine Ecosystems and EEZs, rather than attempting to hindcast time series of global catches (Barange et al. 2014 Nat. Clim. Change, 4(3): 211-216. doi:10.1038/NCLIMATE2119; Merino et al. 2012 Global Environmental Change, 22 (4), 795-806). The linkage to the human component in those studies is also included but is one-way, rather than explicitly being coupled to the natural system enabling the feedbacks between them to be captured - eg. a major advancement in your study. Furthermore, yours is novel because it is the first global application of its kind to predict long-term temporal changes in global fisheries, that successfully reconstructs observed past changes through time, through calibration of the model with data (before doing the projections)) and importantly through explicit consideration of the effects of technological change.

We very much appreciate these suggestions, and have rephrased this sentence as:

'The model builds on prior works that took regional^(Fulton 2010) or unidirectional coupling approaches^(Barange, Merino et al. 2014, Christensen, Coll et al. 2015), by introducing a comprehensive two-way coupling of human and natural components of the system, using relatively simple but well-founded predictive principles applicable to multi-decadal timescales.'

In terms of recent work on the importance of changes in technological efficiency, this paper might also be useful: <http://icesjms.oxfordjournals.org/content/73/4/1226.full>

We thank the reviewer for pointing out this highly relevant paper. We were pleased to see that this in-depth analysis of the Western Pacific purse-seine tuna fleets also shows a rate of technological progress of about 4% between 1993 and 2010, agreeing quite well with our central estimate and other prior works. We have included a citation of the paper.

Page 3 - Figure 1 caption: the model ensemble is mentioned here but it isn't until you look at the SI that you realise this is a single model ensemble (e.g. models form different parameter sets rather than for example different model structures or from different climate model inputs). This could be more clearly stated by adding a sentence, perhaps after at the top of page 4, after the calibration text. Just a minor typo, but the reference numbers aren't in subscript in the caption.

The following sentence has been added, as recommended:

'We show results for an ensemble of five different combinations of parameter values.'

Page 4 - Line 1 - it is not clear what kind of approach for predicting changes in growth is used? Looking at the SI and the additional model documentation in a different paper, the main equation is McKendrick von Foerster but it differs from the trait-based and size-structured food web model of Hartvig et al. 2011 (J. Theor. Biol. 272(1): 113-122. doi:10.1016/j.jtbi.2010. 12.006) which has food dependent growth etc. More detail on the approach should be provided, even if briefly so that readers understand the model when reading this paper. All of the model equations and parameters (with definitions) should ideally be included in the SI (perhaps these could be confined to tables to prevent using too much space) with enough info in the main text to get the gist. I commend the authors for also making everything available on GitHub, but some readers won't go to the extra effort to look at code.

We recognize the importance of this point, and have added the most important model equations to the SI accordingly. We have also added the following text to the sentence regarding growth:

'...by determining the energy available to the trophic web and the metabolic rates of size-structured fish populations.'

Page 4- line 3, same comment as above... more details on how fishing effort was modelled as this isn't clear when consider the main text + SI on their own. providing the equations, including information on how q and E were forced dynamically need to be clearer. Also in the SI perhaps the more descriptive text would be better placed

This has been added to the SI.

Page 4 - also line 3 - re: "poorly constrained parameters" Need to state which parameters were estimated, which ones were not, what are the different parameters. Again model details as suggested above would solve this problem

These details of this have been added to the SI.

Page 4 - perhaps replace "robust optimisation" with optimisation based on a set of model selection criteria (is it history matching?) ... otherwise sounds like it's least squares or Bayesian , but it's not clear

This has been reworded.

Page 4, line 17 - The model is size-structured but I couldn't find any mention of the size selectivity of the gear, which has been explored a lot with size spectrum models (Scott et al. 2014, Methods Ecol. Evol. 5(10): 1121-1125. doi:10.1111/2041-210X.12256, Jacobsen, N.S., Gislason, H., and Andersen, K.H. 2014. Proc. R. Soc. B Biol. Sci. 281(1775): 20132701. doi:10.1098/rspb.2013.2701) including a recent paper that also considers bioeconomic drivers (Andersen, K.H., Brander, K., and Ravn-Jonsen, L. 2015.

Ecol. Appl. 25: 1390-1396. doi: 10.1890/14-1209.1). These are all covered in a recent review by Andersen et al. 2016 (CJFAS, 73(4): 575-588, 10.1139/cjfas-2015-0230). Please can this be clarified somewhere either in the main text or the SI (in general I found the fishing component not that clear but it needs to be as it is crucial to the study).

These details have been added to the SI.

Page 5 line 10, "Given these results, we apply the hindcast..." In this pgh it became unclear whether the pgh above was also the global model hindcast. It seems that it was more about calibration not the complete reconstruction. Perhaps it is better to move this paragraph up, as it is a big main result and cites Figure 1, then report the details of what was needed to achieve it through the calibration (e.g. swap order of the paragraphs?).

We thank the reviewer for pointing out this potential confusion. We have decided to leave the paragraphs in the same order, given that we feel it's important to show the sensitivity-test support for a progress rate of 5% y^{-1} before discussing the standard hindcast, but have added text in order to clarify.

Page 6, line 1 - again interested to know under what types of size selectivity (as size of fish also matters for price) the study assumes, and whether or not it matters...

This has been clarified in the SI; see also the response to Reviewer 2 regarding price.

Page 8, line 3. "The small fish biomass" Up to here and in the SI as far as I can tell the prediction is total biomass across a wide range of sizes (small, medium and large fish) so this sentence is a bit confusing...did you mean to refer to small fish here?

"Small fish biomass" should have read as "Low fish biomass". However we have rewritten this entire paragraph, and this sentence has been removed.

Page 9, line 2 - in terms of highlight outstanding issues for future it would be nice to add here: or feedbacks due to cascading effects of fisheries (Andersen et al 2010), alternative stable states or regime shifts (De Roos et al 2006, Proc Roy Soc B); Horan et al 2011 PNAS)

This is a great suggestion. We have added these thoughts and references.

SI comments:

Page 1-2 -Model description. This is not really enough detail for the paper to be stand-alone. I know it's a lot of detail but even just including a table of equations and parameters would help a huge amount and will mean more people will appreciate what you have done (many won't go to the extra effort of checking other papers or code repositories - although it is excellent that you have done that!). Since it is the important highlight of your method a much clearer description of how time varying effort and

catchability were linked to the natural ecosystem is needed. Some of the details starting on page 5 (pgh 2) would fit nicely at the end of the model description? This would make the historical forcing section, on page 5 much easier to interpret, as it is not fully clear to me how this was implemented in the current text. Overall it would be very helpful if the model equations, parameter definitions and values (highlighting the ones being calibrated and estimated values) were all provided.

We thank the reviewer for these detailed suggestions. The supplementary text has been greatly expanded, accordingly, including equations, parameter definitions, and the parameter values used in the optimization.

Page 4 - "continual transient increases" - what does this mean? Is it a linear increase, exponential increase, extrapolation from an existing time series? More detail would help here.

This has been clarified as:

‘the model [was] integrated through a 200 year transient with increasing technology at 5% y^{-1} , starting from a very low value.’

Page 4 - How good are the model predictions? It was unclear what you meant by saying the ability to predict the peak catches (timing or magnitude for example). Perhaps perhaps this is a wider issue of not being able to assess how good the model performs at the LME scale. Maybe you could include a plot or table showing the differences between the observed and predicted values for each LME or at least reports some aspects of model skill that pertain to the criteria you listed under "Parameter optimization".

We now give the r^2 for each ensemble member across the 55 LMEs (SAUP peaks vs. the model peaks) in the SI, Table S1. We also list the parameter values used for each ensemble member, to show the breadth of parameter values represented in the ensemble.

References cited

- Barange, M., G. Merino, J. L. Blanchard, J. Scholtens, J. Harle, E. H. Allison, J. I. Allen, J. Holt and S. Jennings (2014). "Impacts of climate change on marine ecosystem production in societies dependent on fisheries." Nature Climate Change **4**(3): 211-216.
- Cheung, W. W., T. L. Frölicher, R. G. Asch, M. C. Jones, M. L. Pinsky, G. Reygondeau, K. B. Rodgers, R. R. Rykaczewski, J. L. Sarmiento and C. Stock (2016). "Building confidence in projections of the responses of living marine resources to climate change." ICES Journal of Marine Science: Journal du Conseil: fsv250.
- Christensen, V., M. Coll, J. Buszowski, W. W. Cheung, T. Frölicher, J. Steenbeek, C. A. Stock, R. A. Watson and C. J. Walters (2015). "The global ocean is an ecosystem: simulating marine life and fisheries." Global Ecology and Biogeography **24**(5): 507-517.
- Costello, C., J. Lynham, S. E. Lester and S. D. Gaines (2010). "Economic incentives and global fisheries sustainability." Resource **2**.
- Fulton, E. A. (2010). "Approaches to end-to-end ecosystem models." Journal of Marine Systems **81**(1): 171-183.

Reviewers' comments:

Reviewer #1 (Remarks to the Author):

Many of my concerns have been addressed. And I welcome the inclusion of the MSY case.

Since I'm a resource economist I have emphasized in my first review that not only the biology responds to fishing and/or environmental changes; but fishermen do also respond. The point is not that fishermen respond, but that their response is determined by the regulatory regime. In the paper, open access is assumed which has the implication that fishing effort responds according to the rule that revenue is equal to cost. This determines the fishing pressure and together with environmental factors the development in the harvest and fish stock biomass. If the regulatory regime had been something else then the fishing pressure would be different and hence the development in harvest and fish stock biomass different as well. That's why basic economy forces will be important to include.

The authors indirectly try to accommodate this criticism by finding the MSY level to indicate that this is the highest possible level of harvest (and corresponding fish stock biomass) level from a biological point of view. And they correctly point out that proper management is needed to reach MSY level. However, the MSY level is not the level that produce the maximum economic yield (MEY). And further, as pointed out above, the regulatory regime is not included in the MSY calculation, so the fundamental economic processes are hardly included (except in the open-access case).

Reviewer #2 (Remarks to the Author):

This paper uses a bioeconomic model to reconstruct the history of global fisheries and make projections under different rates of technological change, management systems, and climate change. The recent Costello et al. paper (citation number 17) does similar calculations using a different bioeconomic model, but does not look at technological change nor climate.

I continue to have a number of issues with the basic methods. Even the supplemental materials are incomplete on how the model was actually fit. The SI doesn't actually give the formulas or how fitting to peak catches was balanced with the 8 LMEs. The model appears to predict biomass and catch by the three size groups on a 1 degree grid around the world. There are many ways that this kind of output could be fit to data and my interpretation of the SI is that the dominant criteria is just the peak catch. As the SI notes for those 8 LME's we have both abundance and catch data covering a significant fraction of the global catch - so why not actually fit to those data?

A second issue is the data actually used. Figure 1c refers to reference 6 as a source of effort data. I looked through reference 6 and didn't see effort data. Figure 1e gives reference 2 as the source of abundance data but again I didn't see this in reference 2 (but did see effort

data so perhaps the caption to 1c should refer to reference 2 not reference 6.)

The model fits are simply not convincing - yes they show an increase in effort and a decline in abundance but comparing figure 1e to 1f they look strikingly different. The model predictions show a continued and in fact accelerating decline.

The model scenarios and historical reconstruction have a lot of overlap with reference 17, which also explored scenarios of continued open access and MSY management, and also attempted to reconstruct the history of abundance. I contacted the authors of 17 and obtained a total abundance of harvested species from their analysis shown below.

This figure is shown in the attached pdf.

This is strikingly different from Figure 1f.

As in all statistics, the model would be much better estimated by fitting to contrasting regions and I strongly suggest using regions where catch and abundance data are well documented.

The SI should show the observed and predicted trends in abundance and catch for the 8 specific LME's where good data are available. Indeed I would argue that before the authors attempt to take this model global they should demonstrate that it can replicate the trends seen in different regions.

Reviewer #3 (Remarks to the Author):

The authors have done a great job addressing the reviewer issues and the revised manuscript is substantially stronger as well as being much more transparent and repeatable. I only have a very minor comment: a reference is missing on page 12 of the Supplementary Material in the MSY section.

Perhaps an appropriate reference is: Mace, P.M. and Mace, P. (2001) A new role for MSY in single-species and ecosystem approaches to fisheries stock assessment and management. *Fish and Fisheries*, 2:2-32 or the Larkin 1977 reference it refers to.

Responses to the reviewers' comments are given in italics.

Reviewer #1 (Remarks to the Author):

Many of my concerns have been addressed. And I welcome the inclusion of the MSY case.

We are glad that our revised manuscript addressed many of the prior concerns.

Since I'm a resource economist I have emphasized in my first review that not only the biology responds to fishing and/or environmental changes; but fishermen do also respond. The point is not that fishermen respond, but that their response is determined by the regulatory regime. In the paper, open access is assumed which has the implication that fishing effort responds according to the rule that revenue is equal to cost. This determines the fishing pressure and together with environmental factors the development in the harvest and fish stock biomass. If the regulatory regime had been something else then the fishing pressure would be different and hence the development in harvest and fish stock biomass different as well. That's why basic economy forces will be important to include.

We entirely agree that the response of fishermen to changes in the environment is critical, and this is exactly why our paper is groundbreaking: it is the first to include prognostic representations of both the ecosystem and the humans within a unified framework. Our simulated, interactive changes in fishing effort, coupled to environmental change, provide a clear representation of the most basic economic forces in a comprehensive way that allows long-term simulations. This has not previously been done at the global scale, or on similar timescales, and therefore provides a novel perspective.

We also agree with the reviewer's point that the regulatory regime is critical: indeed, as we point out, regulation will become increasingly important in future. However, rigorously simulating the dynamics of various regulatory regimes in a spatially and temporally explicit model, as the reviewer alludes to, is a major additional undertaking, that goes far beyond the scope of this initial work. We do plan to address this in future, through the ERC-funded BIGSEA project, but it is not a trivial problem.

Our use of the open access dynamic is motivated by the fact that this is quite close to historical fishing, throughout the 20th century (when regulations were largely absent or ineffective) and it is a clearly-defined end member for the future (which is unpredictable). Thus, it is a highly instructive projection as a boundary, although we do not intend to imply that is a likely outcome.

The authors indirectly try to accommodate this criticism by finding the MSY level to indicate that this is the highest possible level of harvest (and corresponding fish stock biomass) level from a biological point of view. And they correctly point out that proper

management is needed to reach MSY level. However, the MSY is level is not the level that produce the maximum economic yield (MEY).

The decision to pursue a target closer to MEY vs. MSY is a societal one, contingent on whether a society chooses economic profit over food security. In practice, most societies are likely to pick something in between. Furthermore, the ability of a society to approach its goal depends on the effectiveness with which the regulatory regime can exert control over fishing effort. Thus, although fisheries economists might often propose MEY as a target, there is no guarantee that the future of the global fishery is more likely to tend towards MEY than MSY or OA - the realized future outcome depends on societal decisions, coupled with the capacity of governance mechanisms to effectively control effort, and will be spatially and temporally variable. These are fascinating topics and well worthy of attention, but are not easily predicted on long timescales, and do not alter our conclusions.

In short, we very much appreciate the enthusiasm of reviewer #1 to push our approach further - and intend to do so in future. However, we do not see any concerns, among these suggestions, that would call into question the robustness or novelty of the present results.

Reviewer #2 (Remarks to the Author):

This paper uses a bioeconomic model to reconstruct the history of global fisheries and make projections under different rates of technological change, management systems, and climate change. The recent Costello et al. paper (citation number 17) does similar calculations using a different bioeconomic model, but does not look at technological change nor climate.

I continue to have a number of issues with the basic methods. Even the supplemental materials are incomplete on how the model was actually fit. The SI doesn't actually give the formulas or how fitting to peak catches was balanced with the 8 LMEs. The model appears to predict biomass and catch by the three size groups on a 1 degree grid around the world. There are many ways that this kind of output could be fit to data and my interpretation of the SI is that the dominant criteria is just the peak catch. As the SI notes for those 8 LME's we have both abundance and catch data covering a significant fraction of the global catch - so why not actually fit to those data?

We can see why the reviewer felt that insufficient detail was provided, as we had mistakenly been under the impression that the more detailed model description paper (Carozza, Bianchi and Galbraith, in revision) would be provided to the reviewers. We appreciate the Reviewer's insistence on this point, as we recognize the value in also having a more fully documented description of the parameter optimization included here. As a result, we have significantly expanded the information provided in the supplemental material.

A second issue is the data actually used. Figure 1c refers to reference 6 as a source of effort data. I looked through reference 6 and didn't see effort data. Figure 1e gives reference 2 as the source of abundance data but again I didn't see this in reference 2 (but did see effort data so perhaps the caption to 1c should refer to reference 2 not reference 6.)

We apologize for the mis-stated reference. Indeed, the citation should have been to Watson et al., reference 2. Other than this mistaken citation number, we are not aware of any problems with the data.

The model fits are simply not convincing - yes they show an increase in effort and a decline in abundance but comparing figure 1e to 1f they look strikingly different. The model predictions show a continued and in fact accelerating decline.

The Reviewer would appear to be focusing on the divergence between the model and observations following approximately 1990, the time period during which management was growing more effective, particularly in countries with stock assessments. Prior to this, the simulated magnitude of relative change in biomass agrees extremely well with the observations, from ~0.8 or 0.9 in 1950 to ~0.4 in 1990.

We would emphasize that the agreement over the earlier period is an emergent property of the model, based on the ecosystem model construction and the parameter optimization, and was not tuned for relative biomass depletion in any way. The subsequent divergence of the time series at the end is consistent with an increasing effectiveness of regulation over the last 20 years, which is not simulated by the open access solution, and does not in any way invalidate the model.

Although we pointed this discrepancy out in the previous version, we realize that the original version of our text may have given the reviewer the mistaken impression that we were arguing that open access explains the entire history of the global fishery, and will determine its future. This is, of course, false - many regions of the world ocean do have effective management regimes in place, aimed at reducing effort. However, open access was likely a good approximation for most of the world up until at least 1980, and may have been reasonably good until 2000.

We initially made only a brief mention of the recent divergence of assessed biomass from the OA model, and given the reviewer's comments, agree that it should be more prominently highlighted. We have reworded the text significantly to emphasize the fact that our results are consistent with a divergence of the world from OA around 1990, and to differentiate this from the reasoning behind our hindcast attribution approach.

Related to this, we have also moved the historical hindcast discussion from the SI to the main text, in order to assure that this important point comes across clearly.

The model scenarios and historical reconstruction have a lot of overlap with reference 17, which also explored scenarios of continued open access and MSY management, and also attempted to reconstruct the history of abundance.

Although reference 17 does address similar topics, and is highly complementary to our work, it does not address the roles of technology or climate change, as the reviewer points out. Since these our results are primarily concerned with climate and technology, we do not see an undue amount of overlap.

I contacted the authors of 17 and obtained a total abundance of harvested species from their analysis shown below.

This figure is shown in the attached pdf.

This is strikingly different from Figure 1f.

The two plots are compared directly here. Note that, since the data we plot were given as a relative biomass fraction compared to estimates of unharvested biomass, our curve has a different y-axis. Because the reviewer did not provide a corresponding estimate of the unharvested biomass, it was necessary to make a linear transform of the y-axis of our plot in order to compare them. Since a linear transform requires two tie-points, we chose the zero-point (which has no uncertainty) and the value over the last 5 years of the shorter timeseries (when presumably the data are most reliable, and the two should therefore agree).

As shown, the appropriately-scaled curves overlap within the error bars for almost the entire period 1960-2006. We would not characterize this as 'strikingly different'. The only discrepancy is at the beginning of the timeseries, when the Reviewer's figure shows a

long period of stability. Given that industrial fisheries were expanding rapidly at this point in history, with negligible management, and tended to target more vulnerable species first (e.g. Srinivasan et al., Bioeconomics 2010), it is hard to imagine why biomass would have stayed constant over this 15-year period.

The source of the biomasses used in the reviewer's plot is not obvious from the Costello et al. paper itself (no similar plot appears there, and the appendix does not provide the relevant details). We suspect that the stock assessment data was not thoroughly corrected for changing coverage and/or other biases in data availability, which is particularly important in the early part of the timeseries when stock assessments were rare. Thus, although we appreciate the reviewer's efforts in investigating this, we do not see any reason to believe the reviewer's plot to be an improvement over the previously published Worm and Branch (2012) data used in our plot. (Of course, we would certainly be happy to use better data if it can be clearly shown that better data exist, including a careful documentation of why they are better.)

As in all statistics, the model would be much better estimated by fitting to contrasting regions and I strongly suggest using regions where catch and abundance data are well documented.

We entirely agree that fitting to contrasting regions is of key importance. That is why, as now explained more clearly in the SI, we use all available regions in the world ocean, other than a few arctic regions where the catch is negligible.

We would reiterate that the use of the peak harvest in each of the LMEs is motivated by the fact that this property is dominated by ecosystem characteristics. This then allows us to use the time-series as independent attribution tests. Our use of an ensemble is precisely to address the fact that there are many ways to fit the data with different parameter combinations - this is one of the key strengths of our approach.

The SI should show the observed and predicted trends in abundance and catch for the 8 specific LME's where good data are available. Indeed I would argue that before the authors attempt to take this model global they should demonstrate that it can replicate the trends seen in different regions.

Again, there was apparently a misunderstanding - we use 55 LMEs, not just 8. We agree that this regional comparison is a critical aspect ensuring the robustness of the model.

However, if the reviewer is suggesting that the model should be able to accurately simulate the temporal stock assessment trends at the 8 best-assessed LMEs over recent decades, we disagree, for three reasons. First, we are simulating all commercial stocks, whereas only a subset of stocks are assessed at any given LME. Second, the LMEs with the most thorough stock assessments are also the best managed - we would not expect to correctly simulate the temporal trends of a well-managed LME with an open access assumption, but would need to introduce arbitrary management

parameterizations, which is inconsistent with our first-principles approach. Third, individual LMEs can be strongly influenced by idiosyncratic local economic factors that are smoothed out by large-scale averaging in the global average trends; this is obvious in comparing the harvest timeseries in individual LMEs with the much smoother global historical trend in harvest.

Nonetheless, we entirely agree that there is much to be gained by taking a close look at individual ecosystems. In fact, we have already embarked on one such study for the Scotian shelf, where we will be making very detailed comparisons of the model with highly-resolved catch, effort and abundance data. As can be seen from the preliminary figure (attached), the same model ensemble used in our manuscript captures the magnitude of the estimated peak well with a simple 5 % y^{-1} increase in technology, indicating a robust performance of the ecosystem model. The prediction of a peak duration of roughly 50 years, rather than 30 years, is consistent with the rapid increase of technology in the Scotian shelf region that occurred during the 1960s, compared with the long-term global average.

Although this type of in-depth comparison is a major undertaking, beyond the scope of the present work, it will ultimately allow a powerful and nuanced new way to explore the time-evolution of local economic forces.

Reviewer #3 (Remarks to the Author):

The authors have done a great job addressing the reviewer issues and the revised manuscript is substantially stronger as well as being much more transparent and repeatable. I only have a very minor comment: a reference is missing on page 12 of the Supplementary Material in the MSY section.

Perhaps an appropriate reference is: Mace, P.M. and Mace, P. (2001) A new role for MSY in single-species and ecosystem approaches to fisheries stock assessment and management. *Fish and Fisheries*,2:2-32 or the Larkin 1977 reference it refers to.

We thank the reviewer for their supportive comments and suggestions - we agree, these are both marvellous papers, and the citations have been added.

Preliminary BOATS simulation for Scotian Shelf LME, using ROMS (unpublished)

ROMS primary production

BOATS harvest at LME peak

Scotian Shelf ensemble mean total harvest and s.d., using 5% y^{-1} technological progress, showing a peak of 2 Mt y^{-1} . N.B. this ensemble includes the same 5 parameter combinations as the *N. Comms* manuscript.

SAUP harvest reconstruction, showing a peak near 1.9 Mt y^{-1} . Note the more shorter timescale in the SAUP reconstruction, consistent with a rapid deployment of technology as international industrial trawlers arrived in the region in the 1960s.

REVIEWERS' COMMENTS:

Reviewer #2 (Remarks to the Author):

The essential message of this paper is that the increase in harvest efficiency of fishing gear when combined with a bioeconomic model can explain the increase and peak in world catches, and then forecast a decline if open access continues.

Certainly there have been some major increases in fishing technology over the last 70 years, with two major jumps occurring with the development of distant water factory trawlers for demersal species and large rapidly sinking nets for high seas tunas fisheries for skipjack and yellowfin that enabled the tuna seiners from the eastern Pacific to move into the Western Pacific and Indian Ocean. Improvements in GPS, acoustics and satellite ocean conditions certainly have improved catching capacity in some cases.

However, when we look at where the increase in global catch came from between 1950 and 1980 it has been dominated by two events. First was the development of the Peruvian anchoveta fishery. This did not depend on any technical innovation, but was simply the transfer of the same technology (and boats) that had been fishing in the California sardine fishery in the 1950s to Peru.

Second has been the development of large number of trawlers and other vessels in China and much of S.E. Asia. There was no new technology developed, the current fleet in S.E. Asia uses 1950's technology.

Thus when we look at what has actually happened, I would have to conclude that there has not been a 5% increase in gear efficiency, but rather it has been the expansion of old technology to much of the world that explains the major increases in catch. The model fits that are similar to historical catches are thus likely fortuitous rather than causal. I would suggest the major change has been the transfer of technology and industrialization of fisheries around the world that has driven the catch pattern.

Looking specifically at Asia the two big changes have been increases in number of boats and in boat size. Neither of these was an increase in catching power, but rather the investment in more and bigger boats. The estimates of global fishing capacity is typically measured in engine power, thus larger boats does not count as increasing the catchability coefficient in the model used.

In the case of China the best explanation for investment was the growth of the market economy. For Peru and much of Latin America it was movement of capital from other places.

The paper poses no competing hypothesis other than alternative parameters of the same model ... it simply argues that because it can fit the pattern of catch it should be accepted.

Reviewer #3 (Remarks to the Author):

This is the third time I have seen the manuscript and I stand by my original reviews that this a robust and novel study.

It is the first global application of a dynamic hindcast model that couples human ecosystem interactions to economic drivers of fisheries by integrating dynamics for physical, biogeochemical, and size-structured food web processes with technological development of fisheries. This approach is clearly set apart from the paper mentioned by another reviewer (Costello PNAS paper), which was a bioeconomic model applied to stock assessment time series that only form a fraction of global fisheries catches and furthermore did not incorporate the effects of environmental variation or change.

The authors have done a fantastic job of providing the much needed detail on the model, which was my initial critique of the paper and it is repeatable through detailed sup mat and provision of code on github.

RESPONSE TO REFEREES

We thank both reviewers for their time in reviewing our revised manuscript once again. The critical comments of reviewer 2 were especially helpful in pointing out aspects of 'technological progress' that needed to be better explained, and have thereby significantly contributed to a stronger final work.

Reviewer #2 (Remarks to the Author):

The essential message of this paper is that the increase in harvest efficiency of fishing gear when combined with a bioeconomic model can explain the increase and peak in worlds catches, and then forecast a decline if open access continues.

Certainly there have been some major increases in fishing technology over the last 70 years, with two major jumps occurring with the development of distant water factory trawlers for demersal species and large rapidly sinking nets for high seas tunas fisheries for skipjack and yellowfin that enabled the tuna seiners from the eastern Pacific to move into the Western pacific and Indian ocean. Improvements in gps, acoustics and satellite ocean conditions certainly have improved catching capacity in some cases.

We agree that these are important examples. However, there are many more examples of 'embodied' technological improvements, such as the invention of monofilament lines, power blocks, and fish aggregation devices. Although the invention of each one of these improvements was a discrete event, their deployment in fleets of the world was a gradual (and ongoing) process, according to the diffusion of technological access and the availability of capital. In addition, we would point out that further improvements are continually occurring in the 'disembodied' technology of any fishery, as skippers and crew become familiar with new gear and new resources. Squires and Vestergaard (Marine Policy, 2013) provide an excellent review of the overlooked importance of technology in fisheries.

This comment has revealed to us how our prior submission was prone to misinterpretation in this regard, and we have accordingly added a more explicit description of technological progress in lines 316-322 of the revised manuscript.

However, when we look at where the increase in global catch came from between 1950 and 1980 it has been dominated by two events. First was the development of the Peruvian anchoveta fishery. This did not depend on any technical innovation, but was simply the transfer of the same technology (and boats) that had been fishing in the California sardine fishery in the 1950s to Peru.

This comment partly reflects a misunderstanding of our use of the term 'technological progress'. We do not intend this to mean only the invention of new devices, but to also include the deployment of those devices in fleets to new regions, and the skill with which they are used. Thus, the example of Peru would reflect an increase in local technology (through diffusion) when the Californian boats arrived. Also, we would point

out that the Peruvian fishery was only responsible for ~6 Mt y⁻¹ on average between 1960 and 2006 (with a peak in 1970), so it was not a dominant factor in the >80 Mt y⁻¹ increase between 1950 and 2006.

The importance of technology diffusion is now mentioned in lines 316-322.

Second has been the development of large number of trawlers and other vessels in China and much of S.E. Asia. There was no new technology developed, the current fleet in S.E. Asia uses 1950's technology.

Again, this comment reveals a misunderstanding of our intention regarding the phrase 'technological progress': the appearance of trawlers, powered seiners etc. in China and SE Asia reflects an increase in technology relative to the technology that existed there previously. Technological progress has played a key role in industrial Asian fisheries, as discussed by Pomeroy et al. (Marine Policy, 2007) for example. Thus, there is no inconsistency.

Thus when we look at what has actually happened, I would have to conclude that there has not been a 5% increase in gear efficiency, but rather it has been the expansion of old technology to much of the world that explains the major increases in catch. The model fits that are similar to historical catches are thus likely fortuitous rather than causal. I would suggest the major change has been the transfer of technology and industrialization of fisheries around the world that has driven the catch pattern.

This suggestion does not disagree with our conclusions, but it ignores half of the process. The appearance of new technologies in fishing fleets of the world requires two things: invention of the technologies (many were invented between 1950 and 2006) and deployment of those technologies, through the transfer of knowledge and goods, the latter of which is dependent on access to capital. The reviewer is focusing on the importance of the transfer aspect - which we entirely agree with, and which is implicitly included in our model - but the first aspect, invention, is of critical importance as well. Our conceptual framework includes both aspects, invention and deployment, within the catchability term.

Looking specifically at Asia the two big changes have been increases in number of boats and in boat size. Neither of these was an increase in catching power, but rather the investment in more and bigger boats. The estimates of global fishing capacity is typically measured in engine power, thus larger boats does not count as increasing the catchability coefficient in the model used.

We agree that in some local fisheries, increases of boat number and size was a very important factor, reflecting an increase in effort (like the reviewer, we define effort as engine capacity). However in all cases these have been accompanied by other types of technological improvements. Our 5% increase is a rough global long-term average, and is not expected to apply to all fisheries at all points in time.

However it is important to recognize that our model does not account for the importance of hunger as a driver of subsistence fishers, who are not included in the formal international fish economy. This is most significant in south Asia, and would be expected to have contributed to an increase in boats as the population has grown, independent of market forces. We now point out the fact that the model includes only industrial fisheries in lines 89-91 of the revised manuscript.

In the case of China the best explanation for investment was the growth of the market economy. For Peru and much of Latin America it was movement of capital from other places.

We agree that these factors are important - but in both these cases, a primary importance of increased availability of capital was the ability to buy more technologically-advanced boats and gear. Thus, the growth of the market economy and access to capital resulted in higher catchability. We now explicitly mention the importance of access to capital in order to implement technology.

The paper poses no competing hypothesis other than alternative parameters of the same model ... it simply argues that because it can fit the pattern of catch it should be accepted.

We would not argue that anything 'should be accepted', as that would not be scientific. We present a new conceptual framework, formalized as the first fully-prognostic coupled global model of fish and fisheries. We use this model to quantitatively illustrate a number of hypotheses, and compare these with historical data to show that technological development was most likely the dominant factor at the global scale. We would note that this was not our initial expectation, but it was a very clear emergent property once we had constructed the model from basic principles. In addition, we show that a rate of change of 5% is consistent with prior estimates from individual fisheries. We agree that it would be nice to compare this with other suitable models, once they are developed, and look forward to seeing this in future work.

In preparing the final revision we have been particularly careful to ensure the language is clear regarding the outstanding uncertainties, and qualified with appropriate caveats.

Reviewer #3 (Remarks to the Author):

This is the third time I have seen the manuscript and I stand by my original reviews that this is a robust and novel study.

It is the first global application of a dynamic hindcast model that couples human ecosystem interactions to economic drivers of fisheries by integrating dynamics for physical, biogeochemical, and size-structured food web processes with technological development of fisheries. This approach is clearly set apart from the paper mentioned by another reviewer (Costello PNAS paper), which was a bioeconomic model applied to

Response to Referees - Galbraith et al.

stock assessment time series that only form a fraction of global fisheries catches and furthermore did not incorporate the effects of environmental variation or change.

The authors have done a fantastic job of providing the much needed detail on the model, which was my initial critique of the paper and it is repeatable through detailed sup mat and provision of code on github.

We thank reviewer #3 for their consistent support and well-reasoned comments.